# From SQL to Knowledge Graphs: An LLM-Driven Multi-Agent Approach with Data Schema Improvement

## Abstract

RDBMS (Relational Database Management System) databases face several limitations, including slow execution with multi-hop queries and a lack of explainability by graphical interpretations. In contrast, Graph database offers a more intuitive and efficient data schema that performs faster execution on large datasets. Most existing RDBMS conversion pipelines focus on running traditional loading commands and relying on Cypher queries. However, the efficiency of using an LLM to generate an effective graph data schema, significantly reducing the ambiguity of the graph database, remains underexplored in the current research literature. This paper presents a novel algorithm that bridges RDBMS and graph database by using a novel LLM-powered ETL agent to standardize table and column names before saving them to the Data Mart. A Multi-Agent System generates a looping discussion between ETL, Analyzer, and Graph agents to optimize the final design through an iterative process of suggesting and scoring the graph database schema. We ensure that the final graph database meets three criteria before being accepted for data conversion: Accuracy, Groundedness, and Faithfulness. This system demonstrates an effective pipeline to automatically convert a tabular database into a graph database through a comprehensive end-to-end process. Our study highlights notable efficiency in using the converted graph database, which is measured on 1,081 samples of the BFSI dataset across three levels of complexity (easy, medium, and hard). Specifically, CypherAgent achieves an 85.6% accuracy for Q&A tasks using a Graph database, which is 12.12% higher than the accuracy achieved by an SQLAgent on the RDBMS database type PostgreSQL, for all queries. Additionally, the Graph database demonstrates faster performance, reducing latency by approximately 3 times. For easy, medium, and hard queries, the Graph database attains accuracies of 90.43%, 81.98%, and 80.06%, respectively, surpassing the RDBMS database by 17.8%, 4.2%, and 11.0%, respectively.

## 1 Introduction

A graph database is a NoSQL database that uses graph structures to store data as nodes (entities) and edges (relationships), making it ideal for highly interconnected data (Corbellini et al., 2017). This new database has a flexible schema that allows data to be represented with rich semantic meaning through nodes and relationships, similar to a human-conceptualized representation of complex data objects in the real-world application (Candel et al., 2022). Therefore, it can be applied in various real scenarios for multiple industries such as computer science, social networks, e-commerce foundations, supply-chain systems, and medical applications (Besta et al., 2023) where graph database accepts a more natural-language network-structured compared to traditional databases such as relational databases (Schneider et al., 2022; Thushara Sukumar et al., 2024).

Graph databases offer substantial improvements in cost reduction and time efficiency over complex multi-hop traversals (Lee et al., 2024; Do et al., 2022). Unlike relational databases, which frequently face performance bottlenecks and scalability issues with complex, multi-table join operations, graph databases are purpose-built for traversing highly interconnected data. Their native representation of various data domains by

entities as nodes and relationships as directed edges allows traversal algorithms to operate in linear or near-linear complexity, even across deeply connected structures. They are also less memory-intensive compared to relational databases, contributing to their efficiency (Kotiranta et al., 2022b). Graph databases also significantly outperform relational databases in terms of query speed (Sandell et al., 2024; Neumann et al., 2009). This highlights the superior performance and efficiency of graph databases in managing and querying connected data.

Additionally, efficiently handling complex data correlations is vital due to the diversity of topics in these datasets, which require sophisticated query optimization techniques (Nathan et al., 2020). Graph databases, with their inherent ability to manage complex relationships and deliver superior performance, align well with complex queries, especially multi-hop reasoning queries.

Most traditional conversions of relational databases to graph databases are based primarily on graphical query language (GQL) loading commands that cannot interpret the relationships among entities within the targeted output graph. The process of migrating data from relational databases to graph databases is complex and resource-intensive, and currently, there is no standardized or widely accepted solution for this conversion (Porter & Ontman, 2020). Despite their transparent set of loading commands, these drawbacks represent a critical barrier to the broader adoption of graph databases.

We primarily propose a novel Multi-Agent System for automatically transforming an unnormalized structured database into a normalized graph database that leverages the knowledge of the structured data schema to yield a better graph database design with higher accuracy and faster answers. We demonstrate its efficiency on a BFSI dataset from a real financial company that contains many internal relationships between tables. We apply data anonymization and masking technologies to hide sensitive information and ensure data privacy policy.

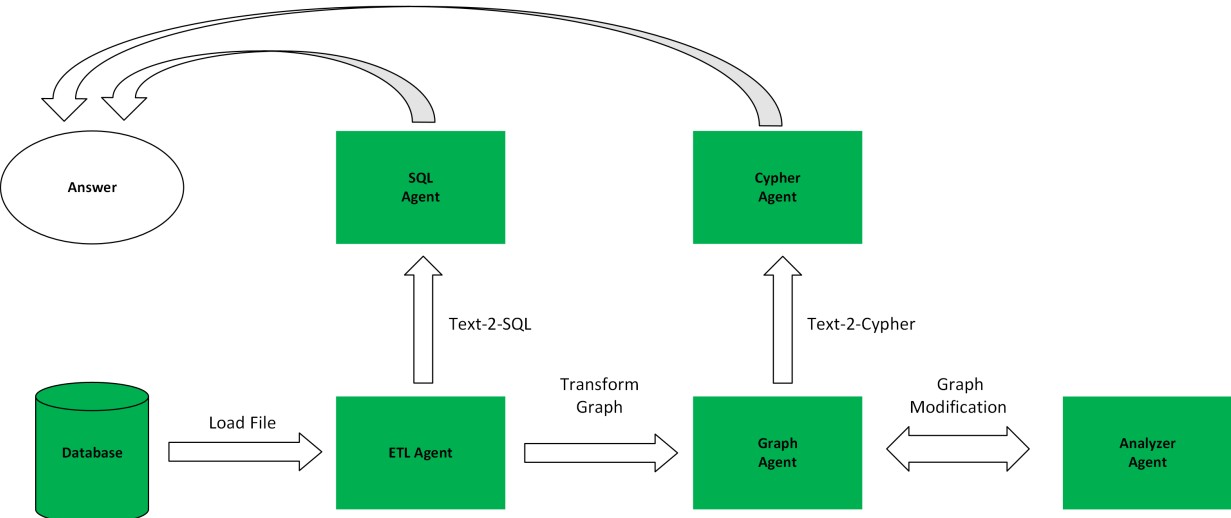

Figure 1: We propose a multi-agent system that realizes an end-to-end question answering (QA) pipeline, spanning from heterogeneous data ingestion to query execution and answer generation. The pipeline ingests data from multiple structured sources and systematically transforms tabular schemas into Knowledge Graph-based representations. Through iterative graph refinement and modification cycles, the system constructs a Standardized Graph Schema that captures real-world entity relationships with explicit semantic alignment and formal graph syntax. To evaluate the effectiveness of the proposed self-evolving graph mechanism, we conduct a comparative study between a Cypher-based agent operating on a Knowledge Graph database and an SQL-based agent operating on a relational database. Experimental results demonstrate that the graph-oriented, multi-agent pipeline achieves superior efficiency and semantic expressiveness in downstream question answering tasks.

We address the complexities and challenges of converting relational databases by leveraging the strength of the LLM-based Multi-Agent System. This closed multi-agent system comprises many specialized agents with particular skills to emulate these distinct roles with the required expertise and collaborate to considerably address complex tasks, especially ETL and database queries. Moreover, this system is an end-to-end solution that can automate and define the knowledge graph structure and ETL data in the same pipeline. To boost the accuracy of Graph database design, we apply a novel Multi-Agent design pattern, which can be reiterated over many refinement turns before reaching the final standard knowledge graph design.

In conclusion, to address the challenges of relational databases, our approach offers several key contributions as follows:

- **Integration of relational and Knowledge Graph databases:** We propose a method that effectively integrates relational databases with Knowledge Graphs, overcoming the limitations of current approaches and enabling seamless data management across different paradigms.
- **End-to-End Data Ingestion Pipeline:** We offer a comprehensive end-to-end data ingestion pipeline that integrates Knowledge Graph architecture with a general Data Ingestion solution, ensuring high accuracy and efficiency in data processing.
- **End-to-End Multi-Agent System:** We introduce an end-to-end Multi-Agent System that produces a round-table discussion with featuring specialized agents with distinct roles: ETL Agent, Analyzer Agent, and Graph Agent. This system automatically designs relationships and nodes for Knowledge Graph architectures.
- **Standard improvement of Knowledge Graph schema procedure:** We establish the first generalized procedure that iteratively evaluates and improves graph structures to be more precise and aligned with the domain-knowledge.

Our approach leverages the cognitive, reasoning, and self-reflective capabilities of Multi-Agent Systems combined with advanced LLM techniques to address existing limitations in database integration. Thanks to automating the design of Knowledge Graph architectures and providing a streamlined data ingestion pipeline that demonstrates a significant advancement in database management and optimization.

## 2 Related Work

**Multi-Agent Systems.** Recent advancements in LLM-based autonomous agents have received significant attention from industry and academia, particularly in software development (Guo et al., 2024). These agent-centric systems, designed to specialize in various coding tasks, have propelled the field forward (Hong et al., 2024; Qian et al., 2023; Chen et al., 2023; Huang et al., 2023; Yang et al., 2024). Typically, these systems assign distinct roles, such as Programmer, Reviewer, and Tester, to individual agents, each responsible for a specific phase of the code generation process, thereby improving the quality and efficiency of software development. In addition, a variety of benchmarks have been developed to assess the performance of these systems in real-world scenarios. For example, SWE-Bench evaluates the ability of multi-agent systems to address actual GitHub issues (Jimenez et al., 2023), while MetaGPT's SoftwareDev provides a comprehensive suite of software requirements from diverse domains, challenging agents to produce fully developed software solutions (Hong et al., 2024). Similarly, (Nguyen et al., 2024) uses Agile Methodology and multiple agents with a Dynamic Code Graph Generator to improve code generation and modifications through updated dependency graphs. In our case, we aim to build upon these advancements by leveraging a smart end-to-end Multi-Agent system to bridge relational databases and Knowledge Graph databases, enhancing data management and integration through advanced LLM techniques.

**Relational vs Graph Databases.** The comparison between SQL and Graph Databases reveals crucial differences in the management of complex and interconnected data. Relational databases, such as MySQL, excel in structured data scenarios with their robust schema enforcement and support for complex queries, including joins and transactions. However, their performance can degrade when handling large, interconnected datasets due to inefficient joins across extensive tables, leading to slower query execution (Tian, 2022; Floratou et al., 2014; Aluko & Sakr, 2019). In contrast, Graph Databases, such as Neo4j, leverage nodes and edges to efficiently model and query interconnected data, significantly reducing the need for complex joins.

This design enables them to handle complex relationships, such as those found in social networks or data provenance systems, with superior performance (Kotiranta et al., 2022a; Cheng et al., 2019; Rodrigues et al., 2023). Graph Databases offer a flexible schema that adapts to evolving data structures, which enhances their efficiency and scalability in dynamic environments (Angles & Gutierrez, 2008; Bonifati et al., 2019). This flexibility reduces data retrieval times and improves scalability, especially in distributed systems. Additionally, transforming relational databases into Knowledge Graph databases can enhance data integration by combining SQL's structured querying capabilities with the flexible, relational modeling capabilities of Graph Databases, as noted in recent advancements (Angles et al., 2023; Chillón et al., 2022; Besta et al., 2023). This approach maximizes the strengths of both paradigms, optimizing data management and integration.

**Data Schema Improvement.** Relational and graph-based data management paradigms differ substantially in how they treat schema: relational databases rely on fixed, table-based schemas, while knowledge graphs support richer semantics, flexible class hierarchies, and dynamic relations, making schema design and evolution in KGs more complex but also more expressive. For relational databases, model-driven approaches to schema evolution, such as EvolveDB, leverage reverse-engineering of existing data dictionaries into abstract models, allow manual schema modifications, compute differences, and automatically derive migration scripts to evolve both schema and data over time (Eckwert et al., 2025). Meanwhile, in the KG context, schema evolution encompasses operations like adding/removing classes, altering class hierarchies, and modifying properties — changes that often require propagating updates throughout the graph to preserve consistency. Bridging the gap between relational data and knowledge graphs has been a major research focus. Some works address this by direct mapping: for example Towards a Complete Direct Mapping From Relational Databases To Property Graphs defines a complete mapping process that transforms any relational database (both schema and data) into a property graph, ensuring information preservation, semantic preservation, and query preservation — and even supports translating SQL queries into equivalent graph-database (e.g. Cypher) queries (Boudaoud et al., 2022). Another recent work, Rel2Graph (Zhao et al., 2023):Automated Mapping From Relational Databases to a Unified Property Knowledge Graph, extends this effort: it automatically constructs a unified property knowledge graph from one or more relational databases and supports the mapping of conjunctive SQL queries into equivalent graph-pattern queries, measuring fidelity with an execution-accuracy metric. Once relational data is mapped into a graph representation, additional techniques address schema optimization for storage and query performance. For instance, (Papastefanatos et al., 2022) proposes a method to extract of properties describing different classes of RDF instances, exploit their hierarchy, and merge/reduce tables to improve performance in an RDB-backed RDF engine, improving both storage efficiency and query performance. Although existing schema improvement algorithms provide systematic ways of mapping between data schema and relational and graph databases. They overly depend on syntactic rules and pre-defined mapping operators that suffer several limitations. Therefore, it motivates us to create an LLM-powered Multi-Agent System capable of semantically standardizing table names, column names, and relationships.

## 3 Multi-Agent ETL system

The rapid growth of business analytics across multiple industries has created a strong demand for systems that can automate database operation, management, and knowledge discovery tasks traditionally performed by humans. In response, we propose a multi-agent system capable of handling multiple aspects of the database lifecycle, including ETL processes, data standardization, data transformation, and data mining.

Our multi-agent framework incrementally formalizes a knowledge graph schema that explicitly represents complex entity relationships, thereby enhancing semantic transparency and interpretability, while simultaneously enabling efficient query execution. In contrast, traditional relational (SQL) databases, although effective for structured tabular data, suffer from several limitations in analytical workloads, including high computational costs for large-scale joins, implicit and schema-bound relational semantics, and sensitivity to rigid query syntax.

Knowledge graph (KG) databases, by comparison, represent data as interconnected nodes and edges, facilitating efficient graph traversals, semantic enrichment, and scalable handling of complex relational structures.

These properties make KGs a more suitable foundation for intelligent, LLM-powered Multi-Agent database systems.

Therefore, we propose a novel system that can automate many stages in a data operational system. Specifically, our detailed architecture is as follows.

## 3.1 Problem Statement

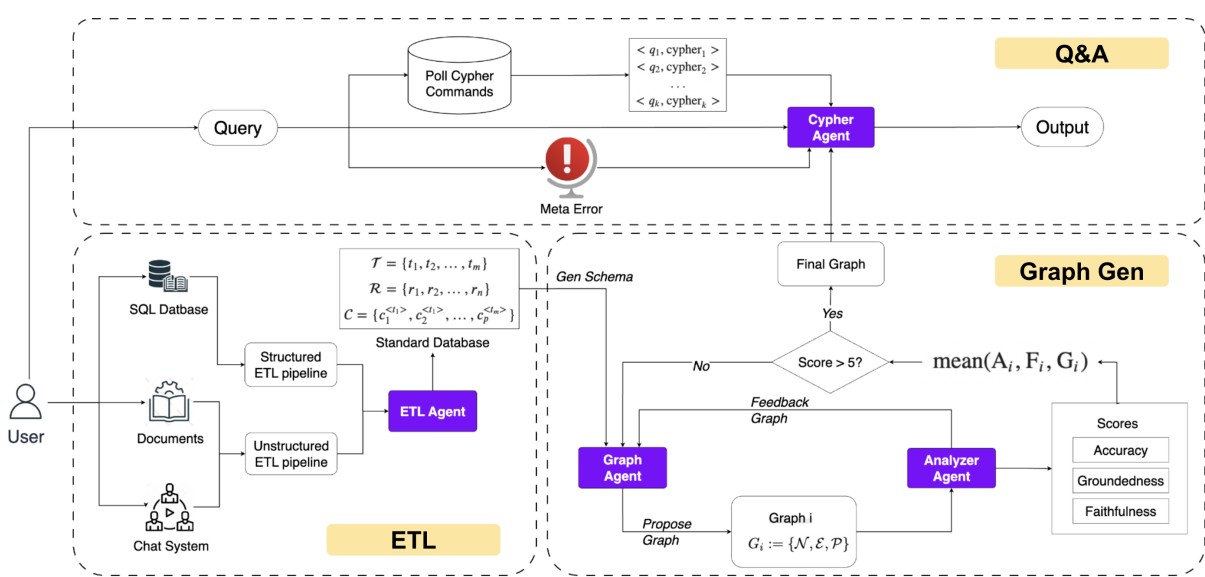

Figure 2: The proposed solution is composed of three primary components. First, the ETL component is responsible for extracting, transforming, and loading both structured and unstructured data into a standardized database through dedicated Structured ETL pipelines. Second, the GraphGen component iteratively evaluates and refines the graph schema design to construct a final knowledge graph optimized for accuracy, groundedness, and faithfulness. Finally, the Q&A component enables the system to address business queries by generating and executing Cypher commands against the constructed graph.

Our solution leverages the SQL database $D_{\mathrm{standard}}$ and the corresponding schema $S$ to create a Graph Schema. The pipeline of transformations is wrapped up inside the Graph Schema for simplicity.

First, the standard data $D_{\mathrm{standard}}$ are loaded and standardized from structured data: $\mathcal{D}_{\mathrm{SQL}}$, CSV tabular data $\mathcal{D}_{\mathrm{csv}}$; and raw data $\mathcal{D}_{raw}$. The SQL and CSV tables will be normalized to their table names, field names, and field formats, ensuring interpretability after transformation by ETLAgent, which defines a prompt engineering to adjust the user column:

---

**Algorithm 1: ETL and standardize dataset**

---

1: **Input:** Structured data: CSV raw data $\mathcal{D}_{\mathrm{raw}}$, SQL database $\mathcal{D}_{\mathrm{SQL}}(\mathcal{T}, \mathcal{R}, \mathcal{C})$; and raw data $\mathcal{D}_{\mathrm{raw}}$.
2: **Output:** Standard dataset $\mathcal{D}_{\mathrm{standard}}$
3: **function** ELT($\mathcal{D}_{\mathrm{SQL}}, \mathcal{D}_{\mathrm{csv}}, \mathcal{D}_{\mathrm{raw}}$)
4:     $\{\mathcal{T}, \mathcal{R}, \mathcal{C}\}_{\mathrm{SQL}} \leftarrow \mathrm{Load}(\mathcal{D}_{\mathrm{SQL}})$                      ▷ Load table, relation, column from SQL database
5:     $\{\mathcal{T}, \mathcal{C}\}_{\mathrm{CSV}} \leftarrow \mathrm{Load}\,(\mathcal{D}_{\mathrm{csv}})$                              ▷ Load table, column from CSV file
6:     $\{\mathcal{N}, \mathcal{E}, \mathcal{P}\}_{\mathrm{Graph}} \leftarrow \mathrm{Load}\,(\mathcal{D}_{\mathrm{raw}})$                ▷ Define node, edge, properties from raw data

7:    $\mathcal{D}_{\text{standard}} \leftarrow \text{LLM}(\{\mathcal{T}, \mathcal{R}, \mathcal{C}\}, \{\mathcal{N}, \mathcal{E}, \mathcal{P}\})$   ▷ Asking ETLAgent to modify table name, column name, datatype (Fig.3)

8:    **return** $\mathcal{D}_{\text{standard}}$

9: **end function**

---

where $\mathcal{D}_{\text{SQL}}(\mathcal{T}, \mathcal{R}, \mathcal{C})$ is the input SQL database comprising Tables $\mathcal{T}$, Relationships $\mathcal{R}$, and Columns $\mathcal{C}$. and $\mathcal{G} = (\mathcal{N}, \mathcal{E}, \mathcal{P})$ is the fine-grained knowledge graph including nodes $\mathcal{N}$, edges $\mathcal{E}$ and properties $\mathcal{P}$.

In the next stage, we produce a Graph generation pipeline that takes the standardized input from ETL Agent and uses Graph Agent to map tables $T_i \in \mathcal{T}$ to nodes $\mathcal{N}$ and relationships $\mathcal{R}$ to directed edges $\mathcal{E}$, with properties $\mathcal{P}$ capturing metadata. The obtained output is a proposal graph schema $G_i$. However, this graph did not meet the technical specifications, like structures and syntax. Therefore, we process a continuous loop for tuning the Graph Schema, relying on its previous version, until it obtains a score greater than $\tau_{\text{thres}}$.

Formally, the transformation process can be expressed as GraphAgent as in Algorithm 2:

$$\mathcal{A}_{\text{Graph}} : \langle \mathcal{D}_{\text{SQL}}(\mathcal{T}, \mathcal{R}, \mathcal{C}), \mathcal{G}_{i-1}(\mathcal{N}, \mathcal{E}, \mathcal{P}) \rangle \rightarrow \mathcal{G}_i(\mathcal{N}, \mathcal{E}, \mathcal{P}),$$

---

**Algorithm 2: Graph generation pipeline**

---

1: **Input:** Structured data: Table schema descriptions $\mathcal{S}$, SQL database $\mathcal{D}_{\text{SQL}}(\mathcal{T}, \mathcal{R}, \mathcal{C})$, CSV Tabular data $\mathcal{D}_{\text{csv}}$; Raw unstructured data: text, pdf, powerpoint $\mathcal{D}_{\text{raw}}$;

2: **Output:** Final graph knowledge representation $G_{\text{final}}$

3: **procedure** GENGRAPH

4:    $\mathcal{D}_{\text{standard}} \leftarrow \text{ETL}(\mathcal{D}_{\text{SQL}}, \mathcal{D}_{\text{csv}}, \mathcal{D}_{\text{raw}})$

5:    $G_0 := \text{GraphAgent}(\mathcal{D}_{\text{standard}})$

6:    **for** $i \leftarrow 1$ to max_iteration **do**

7:        $G_i \leftarrow \text{GraphAgent}(\mathcal{D}_{\text{standard}}, G_{i-1})$

8:        $\text{score}(G_i) \leftarrow \text{SCOREGRAPH}(G_i)$

9:        **if** $\text{score}(G_i) \geq \tau_{\text{thres}}$ **then**

10:          $G_{\text{final}} \leftarrow G_i$                                          ▷ Get final graph

11:          **break**

12:        **end if**

13:    **end for**

14: **end procedure**

---

To ensure the validity of the generated graph schema across schema theory, knowledge graph modeling principles, and domain-specific business semantics, we assign Analyzer Agent to evaluate the architecture of graph according to three aspects:

- Accuracy: The degree to which the nodes, attributes, and relationships in the Graph Schema correctly represent the structures and constraints of the original SQL database.
- Groundedness: The extent to which the Graph Schema conforms to graph database design principles, including syntactic correctness and semantic appropriateness.
- Faithfulness: The degree to which the Graph Schema faithfully captures the intended business logic and supports downstream business operations without introducing distortion.

---

**Algorithm 3: Knowledge Graph Scoring**

---

1: **Input:** Previous Graph Schema $G_{i-1}\{\mathcal{N}, \mathcal{E}, \mathcal{P}\}$, Standard data $D_{\text{standard}}$
2: **Output:** Knowledge Graph scoring (accuracy, faithfulness, groundedness)
3: **function** SCOREGRAPH($G_i$)
4:     $G_i := \{\mathcal{N}, \mathcal{E}, \mathcal{P}\}$
5:     $(\text{A}_i, \text{F}_i, \text{G}_i) \leftarrow \text{Analyzer}(G_i)$                    ▷ Accuracy, Faithfulness, and Groundedness
6:     $\text{score}(G_i) = \text{mean}(\text{A}_i, \text{F}_i, \text{G}_i)$
7:     **return** $\text{score}(G_i)$
8: **end function**

Heuristically, accuracy verifies that nodes, attributes, and relationships correctly reflect the original SQL schema. This aligns with the correctness of schema transformation and lossless information used in database migration research. Groundedness evaluates whether the schema follows standard graph modeling principles. We restrict the Graph Schema from unnecessary intermediaries, require meaningful relationship types, and ensure semantic clarity. These groundedness checks are equivalent to integrity constraints and schema validity used in the ontology of the original relational database schema. Finally, faithfulness evaluates whether the schema preserves domain business semantic meaning and supports downstream tasks without distortion of important business principles and context.

These three metrics generalize long-standing evaluation dimensions from both relational schema theory, knowledge graph modularity, and business understanding. These metrics are organized in a template of the Analyzer Agent prompt as in Figure 4. Each metric is rated on a 5-point Likert scale. The final score is computed as the average of the three metrics. A schema is accepted if its score is $\geq \tau_{\text{thres}} = 5$. The tuning loop runs for a maximum of 10 iterations.

## 3.2 Overall Architecture

The architecture, as shown in Figure 2, is designed to transform structured SQL data into a Knowledge Graph Database (GraphDB) using a Multi-Agent system. Each agent has a distinct role to play, working collaboratively to ensure efficient and accurate conversion. The transformation process can be modeled as:

$$\mathcal{A}_{\text{total}} = \mathcal{A}_{\text{Graph}} \cup \mathcal{A}_{\text{Analyzer}} \cup \mathcal{A}_{\text{ETL}},$$

where $\mathcal{A}_{\text{total}}$ represents the combined functionality of all agents. The system produces a validated GraphDB ready for advanced querying and analytics.

### 3.2.1 System Prompt Pooling

The System Prompt Pooling component serves as the central coordinator, distributing tasks to specialized agents: Graph Agent, Analyzer Agent, and ETL Agent. It provides prompts $\mathcal{P}_{\text{pool}} = \{p_1, p_2, \ldots, p_m\}$, where each $p_i$ contains instructions and parameters for agent tasks. The prompt assignment function is as follows:

$$\mathcal{P}_{\text{pool}} \rightarrow \{\mathcal{A}_{\text{Graph}}, \mathcal{A}_{\text{Analyzer}}, \mathcal{A}_{\text{ETL}}\},$$

ensuring a precise execution of the data conversion process.

### 3.2.2 Multi-Agent System

We generalize the solution to the conversion problem from relational databases to graph databases into an end-to-end Multi-Agent System comprising a set of specialized agents orchestrated within a closed, heuristic-driven pipeline, as follows:

**ETL Agent** handles data ingestion into the GraphDB: $\mathcal{A}_{\text{ETL}} : \mathcal{D}_{\text{SQL}} \rightarrow \mathcal{G}_{\text{init}}$, by extracting, transforming, and loading data while preserving semantic relationships. The output is an initial Graph Schema and standardized structured tables ready for loading into GraphDB.

**Graph Agent**: Continuously improves GraphDB by interpreting the current graph schema and generating an upgraded version of it. It maps structured tables to nodes and relationships to edges: $\mathcal{A}_{\text{Graph}}$ :

$(\mathcal{T}, \mathcal{R}, \mathcal{C}) \rightarrow (\mathcal{N}, \mathcal{E}, \mathcal{P})$, produces an intermediate graph schema, which requires multiple improving iterations before obtaining the final standard one, and executes migration scripts. The schema description $\mathcal{S}_{\text{Graph}}$ ensures that GraphDB is properly structured for data ingestion step.

**Analyzer Agent**: Communicates jointly with the Graph Agent in a unified loop to evaluate the GraphDB based on its schema, $\mathcal{A}_{\text{Analyzer}} : (\mathcal{S}_{\text{Graph}}, \mathcal{D}_{\text{SQL}}) \rightarrow \text{Score}$, where Score measures trustworthiness and robustness. It proposes the most suitable graph schema and standardizes inconsistencies before data ingestion.

**Cypher Agent**: Translates user queries into GraphDB queries: $\mathcal{A}_{\text{Cypher}} : \mathcal{Q}_{\text{user}} \rightarrow \mathcal{Q}_{\text{Cypher}}$, using schema description $\mathcal{S}_{\text{Graph}}$ to ensure accurate data retrieval and effective user interaction.

This MAS system is designed to ensure a high degree of specialization among the agents, where each agent performs its own distinct task and collaborates closely with the others within a closed end-to-end pipeline. Overall, the end-to-end pipeline consists of three stages: ETL, graph generation, and question answering, as described below.

### 3.2.3 End-to-end Pipeline Orchestration

The end-to-end pipeline of Multi-Agent System is strict and logical, which is intuitively depicted in Figure 2. In general, one complete cycle of this pipeline runs across these sequential steps:

**ETL pipeline**: ETL Agent is responsible for standardizing the raw structural data schema, loading them into Graph Database and proposing an initial Graph Schema for Graph Gen step: The input is a raw relational database with its associated schema description, which are then fed into pipeline:

$$\text{Let } \mathcal{D}_{\text{SQL}} \text{ be the set of raw relational databases,}$$
$$\mathcal{D}_{\text{StandardSQL}} \text{ the set of standardized relational schemas, and}$$
$$\mathcal{G}_{\text{init}} \text{ the set of initial graph schemas.}$$
$$\text{ETL}_{\text{pipeline}}(\mathcal{A}_{\text{ETL}}(\mathcal{D}_{\text{SQL}})) = (\mathcal{D}_{\text{StandardSQL}}, \mathcal{G}_{\text{init}}).$$

**Graph Gen**: The Graph Agent and Analyzer Agent jointly interact to finalize the final standard graph database. They repeatedly execute the following looping:

$$G_0 := \mathcal{G}_{\text{init}}, \qquad D := \mathcal{D}_{\text{StandardSQL}}$$
$$\text{For } i = 0, 1, 2, \cdots :$$
$$G_{i+1} = \mathcal{A}_{\text{Graph}}(D, G_i),$$
$$s_{i+1} = \mathcal{A}_{\text{Analyzer}}(G_{i+1}).$$
$$\text{Let } k := \min\{ i \geq 0 \mid s_i \geq \tau \} \quad (\tau \text{ is the acceptance threshold}).$$
$$\text{Then } \text{GraphGen}_{\text{pipeline}}(D, G_0) := G_k = \mathcal{G}_{\text{final}}.$$

If score is equal 5, get the final Graph Schema is accepted and the standardized database is loaded into the final Graph Schema to produce the final Graph Database, which is readily for the Q&A step. Otherwise, the system generates useful feedback to improve Graph Schema and returns to to generate new Graph Schema.

**Generate Answer**: The Cypher Agent uses its understanding of the graph schema and augmented knowledge to provide the accurate answer. It replies on the knowledge of Graph Database, which derived from the schema knowledge and real data of nodes and edges, to generate Cypher queries. In addition, augmented knowledge sources such as few-shot Cypher examples and meta error corrections are injected to further increase accuracy.

$$\text{Let } Q \text{ be the user query}$$
$$\text{Let } F \text{ be the set of Few-shot examples,}$$
$$\text{Let } M \text{ be the set of Meta Errors,}$$
$$\text{Q\&A}_{\text{pipeline}}(\mathcal{G}_{\text{final}}) := \mathcal{A}_{\text{CypherAgent}}(q, F, M, \mathcal{G}_{\text{final}}).$$

In the following sections, we depict in detail each step of end-to-end pipeline in aspects of data conversion and answer generation.

### 3.3 Iterative Review and Schema Refinement

Most existing approaches for transforming relational schemas into graph databases rely on direct execution or rule-based loading procedures that map tables, columns, and foreign keys into nodes and relationships with minimal post-processing Zhao et al. (2023); Virgilio et al. (2013). While such approaches are efficient, they often overlook semantic validation and structural refinement, leading to graph schemas that insufficiently capture domain semantics and exhibit weak or suboptimal relationship modeling Hogan et al. (2021).

To mitigate these limitations, we introduce an iterative review and schema refinement loop during the graph generation stage. Instead of producing a graph schema in a single pass, the proposed approach repeatedly evaluates and modifies the schema across multiple iterations. This cyclic refinement process enables systematic validation of entity types, attributes, and relationship semantics, allowing the graph schema to progressively converge toward a higher-quality representation.

The iterative refinement loop is designed to satisfy the following criteria:

- **Schema consistency:** Node labels, properties, and relationship types are strictly aligned with the structural and integrity constraints defined in the original relational schema, including primary keys, foreign keys, and functional dependencies.

- **Graph modeling principles:** The resulting graph adheres to established graph database design principles, such as appropriate granularity of nodes, avoidance of redundant relationships, and clear separation between entities and attributes.

- **Semantic and business logic preservation:** Domain semantics and implicit business logic embedded in the relational schema are explicitly represented in the graph model to support downstream analytical tasks, such as graph querying, reasoning, and machine learning.

By iteratively reviewing and refining the graph schema, the final output demonstrates improved structural accuracy, semantic expressiveness, and practical utility compared to the initial graph generated without modification. Empirically, this refinement process results in higher schema-level accuracy and better support for downstream applications, validating the effectiveness of the proposed iterative approach.

### 3.4 Data Conversion

The data conversion process is driven by an ETL engine that integrates PostgreSQL, a processing module, and a large language model. The ETL process is defined as:

$$\text{ETL} = \text{Extract} \circ \text{Transform} \circ \text{Load},$$

optimizing the data flow from SQL to GraphDB. The stages are as follows:

- **Data Extraction:** Extracts data from PostgreSQL into formats like CSV: $\text{Extract} : \mathcal{D}_{\text{SQL}} \to \mathcal{D}_{\text{CSV}}$, ensuring manageable data for processing.
- **Data Processing:** Condition data through cleansing, normalization, and transformation: $\text{Transform} : \mathcal{D}_{\text{CSV}} \to \mathcal{D}_{\text{Graph}}$, aligning data with the graph schema.
- **Schema Generation:** The Graph Agent, with the processing module, generates the schema: $\mathcal{A}_{\text{Graph}} : \mathcal{D}_{\text{Graph}} \to (\mathcal{N}, \mathcal{E}, \mathcal{P})$, reflecting the structure and semantics of the SQL data.
- **Data Importation and Verification:** Imports data into GraphDB and validates it: $\text{Load} : \mathcal{D}_{\text{Graph}} \to \mathcal{G}$, with the Analyzer Agent ensuring integrity via iterative refinement until $\text{Score}(\mathcal{G}) \geq \tau_{\text{thres}}$.

This process produces a validated GraphDB optimized for semantic search, data integration, and AI-driven analytics, with the Cypher Agent that facilitates user queries.

### 3.5 Answer Generation

The answer generation process translates user queries into GraphDB responses. The Graph Agent provides the schema description $\mathcal{S}_{\text{Graph}}$, enabling efficient query execution. The Cypher Agent interprets user queries, i.e., $\mathcal{Q}_{\text{user}} \rightarrow \mathcal{Q}_{\text{Cypher}}$, which are executed against GraphDB to retrieve data, i.e., $\mathcal{Q}_{\text{Cypher}}(\mathcal{G}) \rightarrow \mathcal{R}_{\text{data}}$, where $\mathcal{R}_{\text{data}}$ includes relevant nodes, edges, and properties.

In the next step, Cypher Agent formats the response: $\mathcal{R}_{\text{data}} \rightarrow$ Answer, providing a clear business context to adapt the executed data to the current user conversation. This process ensures that non-technical users can access complex datasets effortlessly and understand the answer even though they have never studied Graph Query Language before. Therefore, it helps maximize the utility of the Knowledge Graph.

To enhance the accuracy of the MAS system, we proceed with several experiments on the Graph Query Language generation module of the Cypher Agent, which includes few-shot learning for augmenting new similar couples of (question, cypher commands) pairs as a hint for answering, whereas Meta Error provides knowledge of the common query errors that should be avoided. The detailed arrangement of them in the prompt template setup is described in Figure 5.

## 4 Dataset

We propose a SQL database comprising many basic tables like Loans, LoanTypes, Transactions, Customers, Accounts, Cards, and Branches for the BFSI field. This database will be used as a foundation to develop our Data conversion and Business Question and Answering with the Multi-Agent system. The SQL database will be converted to a standard Graphical Database. Afterward, they will be used as source databases to evaluate the efficiency of the question and answering task on both SQL and Knowledge Graph.

### 4.1 SQL Database Dataset

The SQL database contains a comprehensive dataset with three versions: small, medium, and large, with relevant sizes of $10^3$, $10^5$, and $10^6$ records. They have the same data schema, which includes 9 tables belonging to the banking sector, containing information related to customers, branches, deposits, deposit types, loans, loan types, credit cards, accounts, and transactions. Each table represents different aspects of the banking data.

Table 1: The meaning description of the SQL table in the BFSI field. The database is simulated with fake data to ensure security and privacy.

| Table Name | Meaning |
|---|---|
| Accounts | The banking accounts of customers |
| Branches | The list of banking branches based on their locations |
| Cards | Credit card numbers |
| Customers | Basic information of customers in account registering form |
| Loans | Loan information |
| Loantypes | The type of loans |
| Deposits | Deposit information |
| DepositTypes | The type of deposits |
| Transactions | Transaction information such as amount and date of each account |

These tables are interconnected through foreign keys that link their IDs to another column in a different table, forming a structured relational model. Examples of relationships include:

- **Customers and Accounts:** A one-to-many relationship in which a customer can have multiple accounts.

- **Accounts and Transactions:** A one-to-many relationship in which an account can have multiple transactions.

To increase the clarity of the graph database, our Knowledge Graph Scoring algorithm repeatedly evaluates and modifies the input SQL schema to meet the quality threshold as Figure 6.

## 4.2 Graphical Database

A graphical database is more advantageous than an SQL database because it can encrypt the actual cross-relationship between many tables by edges with semantic meaning expression. After applying the Graph Agent to transform the data, a Knowledge Graph was created with additional features:

- **Semantic Meaning:** Each relationship in the Knowledge Graph is enriched with semantic meaning, making the connections between records more understandable and descriptive.
- **Visual Representation:** The relationships between records are visually represented on the graph, providing an intuitive understanding of the data structure and interconnections.

The Neo4j graph database consists of 9 nodes corresponding to the tables in the SQL database. These nodes are characterized by several properties (e.g., customer_id, balance, loan_type_id, deposit_type_id,...) and are connected by edges representing the relationships (e.g., "HAS_LOAN_TYPE", "BELONGS_TO_CUSTOMERS") as Figure 7.

Using 'GraphAgent' to suggest a new graph schema iteratively, finally, the Neo4j database becomes clear and meaningful. Compared to the original unstandardized SQL schema, the labels for the nodes and relationships were changed to adapt to business logic. Compared with the standardized SQL version, Knowledge Graph has many overwhelming features as follows:

- **Integrity Preservation:** The Neo4j dataset maintains the integrity of the original SQL data while restructuring them into a graph format.
- **Information Enrichment:** The data in Neo4j is enhanced with natural language explanations of nodes, properties, and edges, making the information more interpretable.
- **Ease of Interpretation:** The Knowledge Graph offers a more accessible interpretation of the data through its visual representation, making complex relationships easier to understand.
- **Query Speed:** Queries in the Neo4j database are faster due to the absence of constraints such as indexing in SQL tables. This is especially beneficial for large datasets, where Neo4j can significantly improve query processing times.

The experiments demonstrated how the Transformed Graphical Database outperforms the SQL database on the same benchmarking dataset for three types of business query levels: hard, medium, and easy.

## 4.3 BFSI Business Queries

We propose a new dataset named BFSI, which consists of 1,081 business queries that comprehensively span the operational aspects of the banking and finance domain. To capture a balanced range of difficulty, queries are segmented into three levels: easy (512 queries), medium (333 queries), and hard (236 queries), based on the complexity of their logical structure and the number of entities involved in the underlying graph database.

- Hard level: queries involve complex business questions that simultaneously reference at least two entities within the graph. These queries are designed to incorporate multiple challenging constraints, advanced filtering conditions, nested aggregations, and intricate calculation functions. They often require the use of multi-level groupings, join operations, and domain-specific computations, making them representative of real-world decision-making tasks faced by financial analysts and managers.
- Medium level: queries also relate to two entities but employ simpler logical constructs. Their focus lies on basic aggregation functions such as SUM, AVG, MIN, and MAX, without the need for sophisticated filtering or multi-step reasoning. These queries reflect common analytical tasks that help banks derive insights into customer behavior, product usage, or account-level summaries.

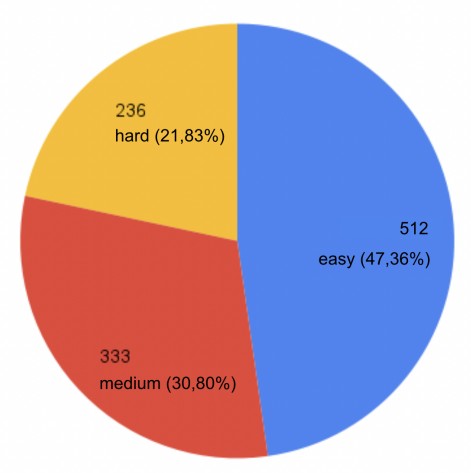

Figure 3: BFSI dataset includes 1081 business queries segmented into hard, medium, and easy levels that cover all business operations of banking and finance.

- Easy level: queries are restricted to a single entity and can be resolved using straightforward query syntax. They usually answer fundamental operational questions, such as entity counts, attribute filtering, or simple comparisons. These queries serve as the foundation for routine reporting and monitoring activities in banking systems.

By organizing the dataset into these three levels, we ensure that it not only reflects the breadth of financial business operations but also provides a structured benchmark for evaluating systems across varying degrees of query complexity.

## 5 Experiments

### 5.1 Experimental Setup

We design a comprehensive evaluation framework to assess the effectiveness of our query generation system across different database paradigms. Our experimental dataset comprises 1089 questions in the banking domain, distributed at three complexity levels with percentages of 47.36%, 30.80% and 21,83% for hard, medium and easy, respectively. These questions cover various banking operations, including transaction analysis, account management, customer relationships, and financial product queries.

The complexity levels are systematically categorized on the basis of query characteristics. Easy queries involve single-entity operations with basic filtering conditions. Medium-complexity queries require two to three relationship traversals across multiple entities. Complex queries feature multiple joins, nested conditions, and aggregate functions throughout the structure of the banking network.

For evaluation metrics, we employ accuracy and latency. The accuracy assessment considers a response correct if it maintains an answer that includes numeric values equivalent to the ground truth, regardless of syntactic variations. This approach ensures fair evaluation when semantically identical answers are expressed differently. We formally define the accuracy score for each question $q_i$ as:

$$A(q_i) = \begin{cases} 1 & \text{if matching}(r_i, g_i) \\ 0 & \text{otherwise} \end{cases} \tag{1}$$

where $\text{matching}(r_i, g_i)$ represents the similar number between the system's response $r_i$ and the ground truth $g_i$. Our implementation utilizes the GPT-4o, GPT-4o-mini, and Qwen3-8B models as the foundation for natural language understanding and query generation.

## 5.2 Setup Evaluation Agents

To process the benchmarking process for the query, we initialize **CypherAgent** and **SQLAgent**. Cypher-Agent handles business queries based on text-2-cypher engine, which converts the input query into the final answer. This agent is aided by a graph database that is migrated from an unstandardized SQL database. On the other hand, SQLAgent is an ReAct SQL Agent that helps answer business queries using the text-2-sql engine. Thanks to the ReAct pattern, this agent is strengthened to Thought, Action, and Observation interleavedly. Another agent called **EvalAgent** is to evaluate the accuracy and latency of two answers from the same query input of CypherAgent and SQLAgent. To keep the equality, we use the same input for both agents, including database schema, relationships, and meta errors. Figure 8 shows the complete pipeline.

## 5.3 Results

We make a comprehensive comparison between the SQL and Cypher approaches at varying levels of query complexity. Our empirical results demonstrate that the Cypher-based approach consistently outperforms the SQL-based method in all evaluation metrics, with particularly notable improvements in both accuracy and computational efficiency, as in Table 3

Table 2: Performance comparison of CypherAgent and SQLAgent variants across different query complexities (*Easy*, *Medium*, *Hard*). CypherAgent incorporates three techniques: meta error (ME), which encodes common query mistakes; few-shot learning (FL) with $k = 3$ examples; and correct error (CE), which analyzes syntax errors and suggests fixes in subsequent rounds. Results are reported in terms of Accuracy (Acc, %), Latency (seconds per sample), and number of Tokens (per sample). CypherAgent-[ME]-[FL]-[CE] with GPT-4o achieves the best overall accuracy while maintaining competitive efficiency, whereas SQLAgent baselines show higher token usage and latency.

| Algorithm | Backbone | Easy | | | Medium | | | Hard | | |
|---|---|---|---|---|---|---|---|---|---|---|
| | | Acc | Latency | Tokens | Acc | Latency | Tokens | Acc | Latency | Tokens |
| CypherAgent-[ME] | GPT4o | 87.70 | 3.19 | 1143 | 77.18 | 2.38 | 1156 | 71.61 | 2.75 | 1177 |
| CypherAgent-[ME]-[FL] | GPT4o | 79.49 | 3.28 | 1309 | 76.28 | 3.81 | 1354 | 61.44 | 3.05 | 1411 |
| CypherAgent-[ME]-[FL]-[CE] | GPT4o | 90.43 | 1.73 | 1314 | 81.98 | 1.94 | 1364 | 80.08 | 2.64 | 1458 |
| CypherAgent-[ME]-[CE] | GPT4o | 80.27 | 1.66 | 1239 | 76.28 | 2.04 | 1285 | 61.02 | 3.23 | 1353 |
| SQLAgent-[ReAct] | GPT4o | 72.66 | 5.63 | 3188 | 77.78 | 6.91 | 2624 | 69.07 | 7.30 | 3284 |
| SQLAgent-[ReAct] | Qwen-8B | 64.70 | 2.17 | 2332 | 58.30 | 2.66 | 2355 | 48.70 | 2.56 | 2385 |
| CypherAgent-[ReAct] | Qwen-8B | 58.40 | 2.13 | 3547 | 46.90 | 2.65 | 3562 | 45.80 | 2.99 | 3578 |

Table 3: Comparison between the best-performing CypherAgent and SQLAgent across different query complexities. CypherAgent consistently outperforms SQLAgent, achieving an average accuracy gain of +12.12% while reducing latency by over 3 times and total tokens by over 2 times. Results are reported in terms of Accuracy (%), Latency (seconds), and relative Accuracy Gain (%).

| Query Complexity | Accuracy (%) | | Latency (seconds) | | Tokens | | Accuracy Gain |
|---|---|---|---|---|---|---|---|
| | Cypher | SQL | Cypher | SQL | Cypher | SQL | (%) |
| Easy | 90.43 | 72.66 | 1.73 | 5.63 | 1,314 | 3,188 | +17.77 |
| Medium | 81.98 | 77.78 | 1.94 | 6.91 | 1,362 | 2,624 | +4.20 |
| Hard | 80.08 | 69.07 | 2.64 | 7.30 | 1,458 | 3,284 | +11.01 |
| Mean | 85.57 | 73.45 | 2.00 | 6.34 | 1,361 | 3,035 | +12.12 |

In the easy query category, which primarily comprises single-entity retrievals and basic filtering operations, Cypher achieves an accuracy of 90.43%, surpassing SQL's 72.66% by a significant margin. This 17.77 percentage point improvement is accompanied by a 3.9 second reduction in latency (1,73 seconds vs. 5,63 seconds), demonstrating the inherent efficiency of graph-based query processing even for straightforward operations.

The performance disparity becomes more pronounced in the medium complexity category, where Cypher maintains its accuracy of 81.98%, while SQL's performance achieved 77.78%. This substantial 4.2 percentage point improvement is particularly noteworthy as it represents queries involving moderate relationship

traversals and multi-entity interactions. The latency differential also widens, with Cypher requiring only 1.94 seconds compared to SQL's 6.91 seconds, highlighting the scalability advantages of graph processing.

For hard queries, which encompass complex multi-hop relationships and sophisticated pattern matching, both approaches exhibit decreased performance. However, Cypher maintains its superiority with 80.08% accuracy compared to SQL's 69.07%. The latency measurements in this category reveal the most significant temporal efficiency gap, with Cypher operating at 2,64 seconds versus SQL's 7.30 seconds, underscoring the graph database's ability to handle complex relationship patterns more efficiently as in Figure 9.

The aggregate results demonstrate Cypher's robust performance advantages, achieving a mean accuracy of 85.57% compared to SQL's 73.45%, with an average latency improvement of 4,39 seconds (corresponding to 68,8% faster) across all complexity levels (Table 3). These findings provide compelling evidence for the efficacy of graph-based approaches in LLM-driven query generation systems, particularly within domains characterized by complex interconnected data structures. The results suggest that graph databases offer a more suitable foundation for natural language query processing in enterprise-scale applications.

# 6  Ablation Study

In addition, the performance comparison of Cypher components over the BFSI dataset demonstrates that Meta Error improves accuracy by 28.84% while greatly reducing token usage, and Few-shot Learning achieves the highest accuracy 85.57%, corresponding to 4,63% improvement, with the lowest latency. Both approaches demonstrate clear efficiency gains over the baseline because the Baseline Cypher usually gets syntax errors that require many retry turns. These errors lead to substantially higher latency and token consumption. Table 4

Table 4: Performance comparison of Baseline Cypher, Meta Error, and Few-shot Learning. Adding meta error can significantly increase the accuracy by 28.84%, whereas, Few-shot learning improves 4.63%.

| Method | Accuracy | Latency (seconds) | Tokens | Accuracy Gain |
|---|---|---|---|---|
| Baseline Cypher | 52.11% | 2.482 | 3,558 | — |
| Meta Error | 80.94% | 2.846 | 1,154 | +28.84% |
| Few-shot Learning | 85.57% | 1.995 | 1,361 | +4.63% |

**Few-shot learning is effective**: Initially, the query system relied on two base SQL Agent for querying relational data in PostgreSQL and Cypher Agent for handling graph-based data in Neo4j. These agents were designed without similar examples. Therefore, it lacks the experience to learn the correct business logic, especially in complex relational databases such as banking and finance. After adding the few-shot examples, it reinforces the business understanding of these agents a lot.

**Meta error adding**: After analyzing the initial version of the Multi-agent system, we realize that it usually commits common types of error in syntax and selects the right tables and columns. Therefore, we store all common types of errors in a meta error database. Adding them to Agent prompt will significantly reduce syntax errors and increase the percentage of runnable code.

In conclusion, our result demonstrates that the full system, combining Meta Error correction with few-shot grounding, substantially outperforms the baseline in both effectiveness (accuracy) and efficiency (token usage, latency). This supports our claim that a judicious integration of common errors and few-shot examples prompting is essential for reliably translating relational queries into correct graph queries in real-world, domain-specific databases.

# 7  Conclusion & Future Work

We propose a novel system that systematically converts structured databases into knowledge graph databases while preserving critical fields and relationships across multiple tables. Our results demonstrate that em-

ploying a multi-agent system yields superior performance compared to a single large language model–based Question Answering system when applied to business queries.

Notably, our proposed solution, CypherAgent, outperforms a traditional SQLAgent in terms of efficiency, as it leverages graph traversal for query answering rather than relying solely on indexing constraints. In addition, CypherAgent achieves higher accuracy by enhancing the semantic representation of relationships across multiple schemas. The overall token usage of CypherAgent is significantly lower than that of SQLAgent due to optimized prompting and a reduced number of reasoning steps. Furthermore, our experiments show that incorporating techniques such as Meta-Error correction and Few-Shot Learning strengthens syntactic robustness by integrating prior knowledge of common errors and domain-specific expertise.

In general, our multi-agent solution establishes an end-to-end pipeline that automates key database operations including normalization, migration, and analysis. This system provides a scalable mechanism for extracting, loading, and transforming structured databases into graph-based representations. Empirical evidence indicates that graph databases are robust and effective in the BFSI domain, with strong potential for generalization across diverse industries and large-scale enterprise environments. These findings highlight the promise of multi-agent knowledge graph systems as a foundation for future intelligent data management and decision-support applications.

In future research, we plan to extend the proposed multi-agent framework by incorporating reinforcement learning and self-reflective reasoning to enable continuous improvement of schema generation and query accuracy. We also aim to generalize the Circle Discussion pattern beyond the BFSI domain to other sectors such as healthcare, e-commerce, and logistics, validating the adaptability of our approach to heterogeneous data ecosystems. Additionally, integrating multimodal data (e.g. text, images, and time-series) into the knowledge graph will further enhance semantic richness and support more complex analytical tasks. Large-scale deployment and benchmarking on real-world enterprise datasets will be conducted to assess scalability, robustness, and end-to-end automation efficiency. Ultimately, this future direction seeks to establish a foundation for fully autonomous data management systems driven by collaborative, reasoning-capable AI agents.

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

## A    Appendix

## B    Appendices

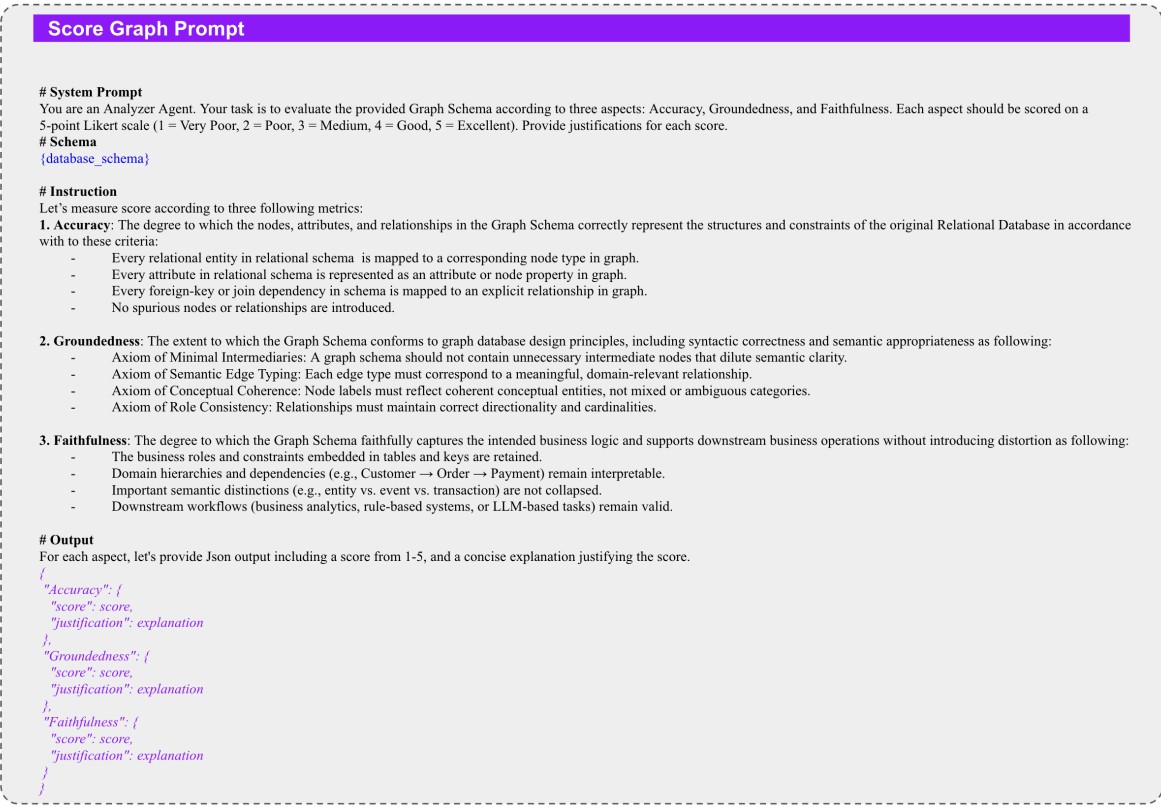

Figure 4: Prompt for scoring graph quality according to three aspects: Accuracy, Groundedness, and Faithfulness. To ensure the foundation of scoring, we require a detailed explanation of why this graph scores the Database schema. Only the design that meets a maximum of 5 for all three metrics must be accepted.

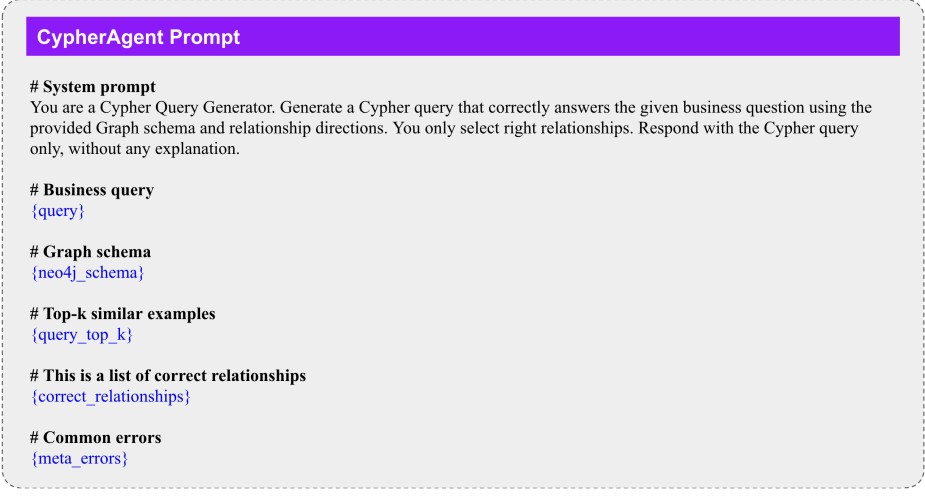

Figure 5: Prompt for CypherAgent that leverages few-shot learning, meta error, and graph schema as the augmented information of the syntax generation process. Thanks to having the common types of errors in meta error, this agent can avoid incorrect syntax. To increase the business logic, the suggested top-k similar examples are used to reinforce its understanding. Finally, the correct relationships are accounted for in order to foster the ability to generate the right relationship directions.

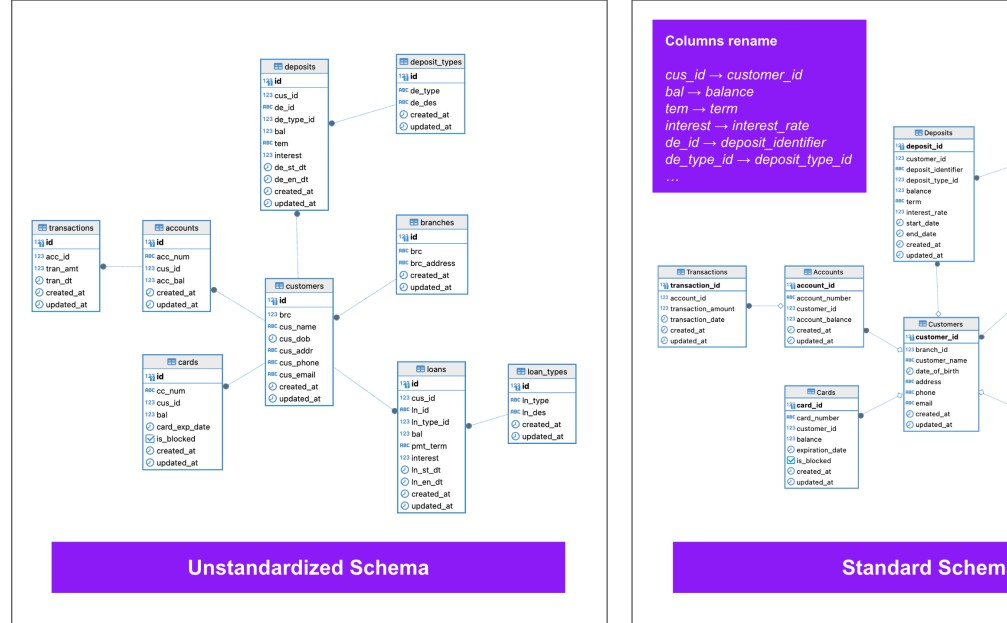

Figure 6: General SQL Database Schema design of BFSI that includes Loans, Loantypes, Transactions, Accounts, Customers, Cards, and Branches with relevant relationships among these tables. On the left is an unstandardized database schema with naming violations like short names and obscure meanings. On the right is a standard schema with the right meaning convention that obtains a maximum of 5 scores for accuracy, groundedness, and faithfulness.

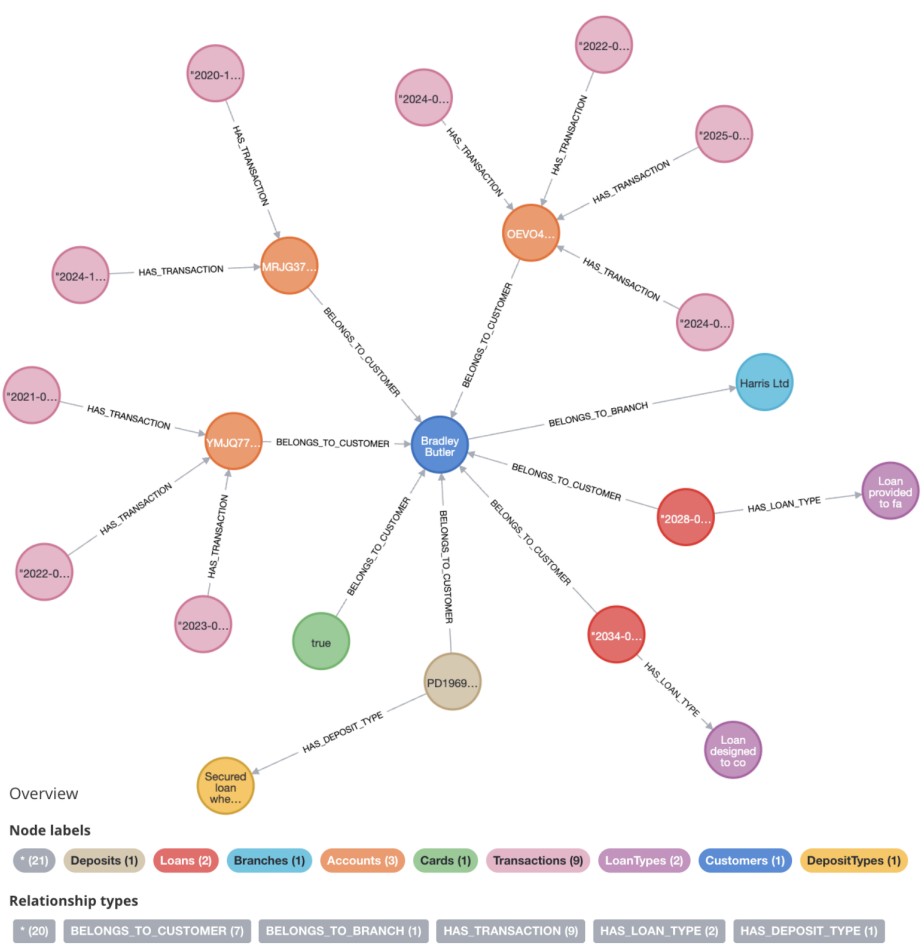

Figure 7: Graphical Database Design is generated by GraphAgent that links multiple entities by edges to simulate the true relationship of the original SQL table. The meaning of each entity is similar to Table 1 descriptions.

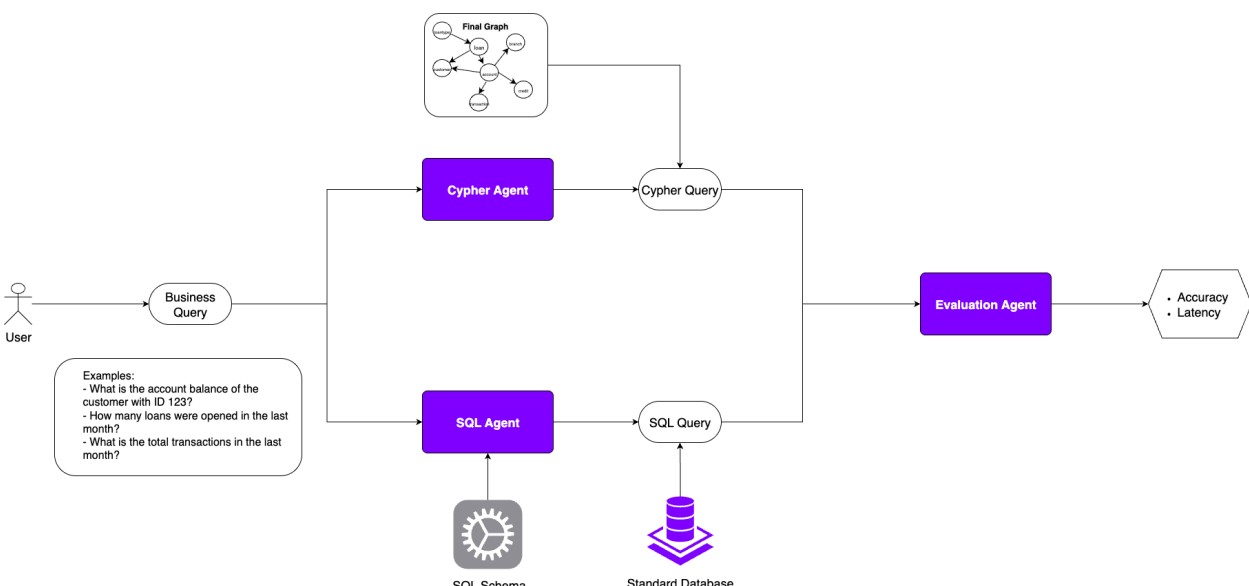

Figure 8: We evaluate Agent over a business dataset by employing SQL Agent and Graph Agent to generate SQL and Cypher queries, respectively. To demonstrate that Graph Agent outperforms SQL Agent, we utilize an Evaluation Agent for automated accuracy and performance measurement.

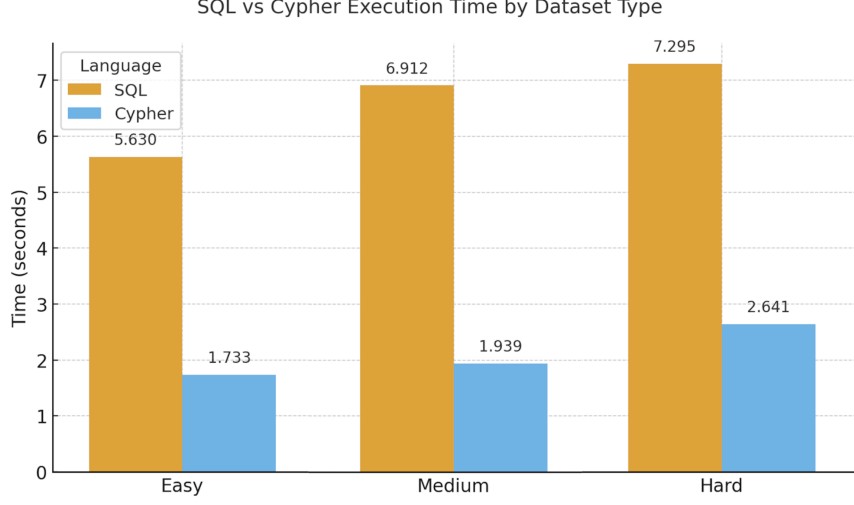

Figure 9: Average Execution time of SQLAgent compared with CypherAgent on three levels: Easy, Medium, and Hard. The plot demonstrates that CypherAgent is faster than SQLAgent because CypherAgent has an optimized Schema, which usually requires a single request to obtain a result. However, SQLAgent uses ReAct pattern, which requires multiple rounds of Thought, Action, and Observation to archive the final answer.

