# OpenReview forum: "From SQL to Knowledge Graphs: An LLM-Driven MultiAgent Approach with Data Schema Improvement"
_TMLR — Rejected by TMLR_

### Review · Reviewer_JfDY · 2026-02-08

**Summary Of Contributions:**

This paper introduces an LLM-driven multi-agent framework that automatically converts relational databases into semantically rich Knowledge Graph databases while improving schema quality and downstream query performance. The authors design an end-to-end pipeline that integrates ETL standardization, iterative graph schema generation, and query answering within a collaborative Multi-Agent System composed of ETL, Graph, and Analyzer agents. The system iteratively refines graph schemas using three principled evaluation criteria—Accuracy, Groundedness, and Faithfulness—to ensure structural correctness and business semantic alignment. Empirically, the resulting graph databases demonstrate superior effectiveness and efficiency, achieving higher question-answering accuracy and significantly lower latency compared to traditional SQL-based querying, validating the benefits of graph-oriented schema design and agentic automation.

**Audience:**

No

**Audience Explanation:**

1. The work primarily integrates existing components (LLMs, ETL pipelines, and multi-agent orchestration) in an engineering manner, with few fundamentally new algorithms or learning principles that would substantially advance the ML research community.
2. The paper focuses on system design and empirical performance without providing formal analysis, guarantees, or deeper understanding of why the proposed multi-agent or schema refinement mechanisms work, which reduces its scientific contribution.
3. The study is centered on a domain-specific database conversion and BFSI workload, making the findings more relevant to applied data engineering or industry use cases rather than broadly impactful to the general TMLR machine learning audience.

**Claims And Evidence:**

No

**Claims Explanation:**

1. The evaluation is conducted only on a single domain-specific BFSI dataset, lacking validation across multiple databases or domains, which makes it difficult to assess the robustness and generalizability of the proposed multi-agent pipeline.
2. The comparison mainly involves a SQLAgent with prompting-based ReAct strategies, without including stronger or more optimized relational baselines (e.g., tuned SQL engines, schema-aware query planners, or hybrid SQL–graph systems), which weakens the claim that the performance gains are intrinsic to the graph-based approach.
3. Although accuracy and latency gains are reported, the paper does not provide sufficient ablations or controlled studies to isolate the contributions of each component (ETL standardization, multi-agent discussion, schema scoring, few-shot/meta-error), making it unclear which module is responsible for the improvements and whether the gains are statistically significant.

**Requested Changes:**

1. The current experiments are limited to a single BFSI dataset. To support claims of generality and robustness, the authors should validate the framework on additional databases from different domains (e.g., healthcare, e-commerce, or public benchmarks) and demonstrate consistent gains.
2. The paper should compare against more competitive relational and hybrid approaches, such as optimized SQL engines, schema-aware query planners, or existing relational-to-graph conversion systems. Additionally, compute budgets and prompting strategies should be normalized to ensure fairness.
3. Conduct systematic ablations to isolate the contributions of each module (ETL standardization, multi-agent discussion, scoring mechanism, few-shot/meta-error prompts). This would clarify which design choices drive performance improvements and strengthen the paper’s scientific insights.

---

> ### Author Response · Authors · 2026-03-08
> **Author rebuttal from SQL to Knowledge Graph: An LLM-Driven MultAgent approach with Data Schema Improvement**
>
> We sincerely thank the reviewer for the thoughtful and detailed evaluation of our paper. Below, we provide a structured response to each major concern raised.
>
> 1. Evaluation Limited to a Single BFSI Dataset
> "The evaluation is conducted only on a single domain-specific BFSI dataset, lacking validation across multiple databases or domains, which makes it difficult to assess the robustness and generalizability of the proposed multi-agent pipeline."
>
> --> We respectfully argue that the primary novelty of our work is not benchmark-driven model ranking, but rather a pipeline for schema improvement of unnormalized databases. This distinction fundamentally determines which datasets are appropriate for evaluation. Existing standard benchmarks such as Spider 2.0 are designed to rank LLM-based Text2SQL models across diverse, pre-structured schemas. By design, these datasets begin with well-formed, semantically clean relational schemas. As such, they are inherently not suited for evaluating the capability our pipeline addresses: the detection and iterative remediation of schema level deficiencies (e.g., ambiguous column names, missing relational semantics, unnormalized table structures).
>
> 2. Weak Relational Baseline Comparison
> "The comparison mainly involves a SQLAgent with prompting-based ReAct strategies, without including stronger or more optimized relational baselines (e.g., tuned SQL engines, schema-aware query planners, or hybrid SQL- graph systems), which weakens the claim that the performance gains are intrinsic to the graph-based approach."
>
> --> We appreciate this concern and wish to clarify the scope of our experimental comparison. The central claim of our paper is not that graph databases universally outperform all SQL configurations, but rather that our end-to-end MAS pipeline combining schema improvement, Knowledge Graph generation, and CypherAgent querying, achieves superior accuracy and efficiency compared to the equivalent SQL-based workflow for the same queries and same LLM backbone. The SQLAgent baseline we employ uses the ReAct prompting pattern, which represents the current state-of-the-art paradigm for LLM-based SQL query execution. ReAct is the dominant design adopted in industry and research for agentic Text2SQL systems, and normalizing both agents to use the same LLM ensures a fair comparison of the database paradigm.
> To ensure a fair and normalized comparison:
> - Both CypherAgent and SQLAgent receive the same input (database schema, relationships, meta-error prompts).
> - Both agents are evaluated on identical 1,081 questions across three complexity levels.
>
> 3. Insufficient Ablation Study
> "The paper does not provide sufficient ablations or controlled studies to isolate the contributions of each component (ETL standardization, multi-agent discussion, schema scoring, few-shot/meta-error), making it unclear which module is responsible for the improvements."
>
> --> We respectfully note that Section 6 of our paper presents an ablation study on the CypherAgent components, and the results provide clear, quantitative evidence of each component's contribution. Specifically, These results clearly demonstrate that Meta Error correction delivers the dominant accuracy improvement (+28.84%), while Few-shot Learning provides additional gains (+4.63%) with a meaningful reduction in latency. Regarding the broader MAS components (ETL standardization and multi-agent schema scoring), we note the following: ETL Standardization, the contribution of this component is demonstrated qualitatively through Figure 6 in the paper, which contrasts the unstandardized schema (with abbreviated, ambiguous column names like 'cus_id', 'tran_at', 'bal'). The improvement in schema directly enables the graph generation step to produce semantically meaningful node and relationship labels.  The comparative performance of CypherAgent on the standardized graph (85.57%) versus the SQL agent on the standardized SQL schema (73.45%) implicitly.
>
> 4. Scientific Novelty and Broader ML Relevance
> "The work primarily integrates existing components (LLMs, ETL pipelines, and multi-agent orchestration) in an engineering manner, with few fundamentally new algorithms or learning principles."
>
> --> We respectfully challenge the characterization of our work as purely engineering integration. Our paper introduces several principled algorithmic contributions that go beyond component assembly:
>
> - Principled schema evaluation framework: We introduce a novel three-axis evaluation framework (Accuracy, Groundedness, Faithfulness) grounded in both relational schema theory and KG modeling principles. This is not a standard engineering pattern but a formalized evaluation ontology for schema quality enabling the MAS to self-regulate its outputs through a principled scoring loop.
>
> - Meta-Error Prompting as a Learning Mechanism: The CypherAgent's integration of meta-error prompting injecting a structured knowledge base of historically observed syntax errors into the prompt

---

### Review · Reviewer_dbUN · 2026-02-28

**Summary Of Contributions:**

The paper introduces an LLM-powered multi-agent system to standardize relational database schema and iteratively generate and refine a knowledge graph schema. They evaluate the proposed system on the VFSI dataset containing 1081 natural-language queries segmented into three levels: easy, medium, and hard. The experiments compare a graph-based CypherAgent against a relational SQLAgent using GPT-4o, GPT-4o-mini, and Qwen-8B backends and show that the CycleAgent achieves a mean gain of 12.12% along with a three-fold latency reduction.

Strengths:
1. The research is timely, using LLM agents for data management pipelines
2. The proposed MAS is end-to-end and the paper provides a fairly detailed architectural diagram and pseudo-code for the ETL, graph-generation, and QA components.
3. The experiments try LLMs of different scales including a bigger GPT-4o model as well as a small GPT-4o-mini, as well as an open-weights model that helps with a sense of scalability
4. The ablation of prompt enhancements is also informing of the measurable gains

Weaknesses:
1. The paper does not discuss some seminal works around the central hypothesis that graph-based QA outperforms relation QA. The paper should also compare the proposed method with some standard W3C recommended methods for the direct mapping of relational data to RDF like R2RML, RDB2RDF, Turtle syntax etc.
2. The dataset generation process is also not described fully - how were the queries authored? how were the ground truth obtained and if they were manually verified? This would directly affect the reliability of the accuracy metric reported in the paper.
3. The claim that this is the "first generalized procedure" for evaluating graph structures is inaccurate given the existence of the RAGAS framework and previous research on semantic and information preservation in database migration.

**Additional Comments:**

- How were the 1081 natural‑language queries generated? Please provide inter‑annotator agreement if manual.
- Have you evaluated the MAS against a single‑LLM baseline that performs the entire pipeline without agent decomposition? If so, how does its performance compare?
- What hardware and software environment were used to measure latency? Are the reported times per query or per batch?
- Did you perform multiple random seeds (e.g., 5 runs) for each configuration? If not, can you provide variance estimates?
- How does the system handle schema evolution (e.g., new tables added after initial conversion)?

**Audience:**

Yes

**Audience Explanation:**

The use of a multi-agent "circle discussion" for iterative schema refinement is a timely topic. However, the interest is currently limited by the paper’s failure to situate itself within the existing literature of agentic self-reflection and W3C data standards.

**Claims And Evidence:**

No

**Claims Explanation:**

- Claiming that graph databases are more suitable for natural-language query processing in enterprise-scale applications based on a single (synthetic) benchmark seems a bit of a stretch. Furthermore, the faithfulness and groundedness of the generated schema are only supported by a 5-point likert scale performed by the same analyzer agent (was some independent human validation performed?)
- Baselines are limited to a ReAct-style SQL agent, and no copmarison to existing automated schema-conversion tools is included.

**Requested Changes:**

1. Include at least one publicly available relational dataset and a corresponding KG conversion to demonstrate that results are not dataset‑specific.
2. Compare against state‑of‑the‑art relational‑to‑graph mapping tools and against a non‑MAS LLM pipeline to isolate the contribution of the multi‑agent orchestration.
3. Define precisely how “accuracy” handles multiple valid answers, rounding, and partial matches; also detail latency measurement conditions.
4. Add a brief statement describing data generation, anonymization steps, and any ethics board approval (even if not required)
5. Is it possible to release the full query set, the ground‑truth answers, and the prompt templates (including the meta‑error database) to enable replication?
6. Is it possible to provide human evaluation of the generated KG schema (e.g., rating by a database engineer) to substantiate the “accuracy/groundedness/faithfulness” scores?

---

> ### Author Response · Authors · 2026-03-08
>
> We sincerely thank you your feedback. It helps us a lot to improve our research. Below we clarify several points regarding the design objectives and evaluation methodology of our work.
>
> 1. On Dataset Evaluation (Standard Text2SQL Benchmarks)
> You suggests comparison against standard Text2SQL datasets such as Spider 2.0. However, this conflates two fundamentally different evaluation targets. Standard Text2SQL benchmarks like Spider 2.0 are designed to rank LLM models under a fixed, well-structured schema, they assume the relational schema is already normalized and optimized. Our system's core contribution is precisely the schema improvement pipeline: starting from a raw, improperly structured schema (with denormalized columns, missing relationships, and ambiguous table semantics), our MAS iteratively transforms it into a graph-ready structure.
>
> 2. On Generalizability of the Pipeline
> You characterizes our pipeline as "fixed" and not a generalized approach. We respectfully disagree. Our system addresses a general three-phase problem: (1) ETL standardization, (2) Knowledge Graph schema generation, and (3) natural-language Q&A over graph databases. Each phase adheres to industry data execution standards and is designed to be modular. The ETL component handles arbitrary relational schemas; the Graph Generation module applies iterative LLM-driven refinement loops that are schema-agnostic; and the Cypher Agent is generalizable to any Neo4j-compatible graph. The BFSI instantiation is one domain application of a general architecture the pipeline is not coupled to financial data.
>
> 3. On the BFSI Dataset and Industry Relevance
> You do not explicitly criticize the dataset, but do request a publicly available dataset for reproducibility. We acknowledge this concern and note that the BFSI dataset serves a dual purpose: it advances academic research by providing a benchmark in an underserved but high-impact domain (banking and financial services), and it narrows the gap between research and industry deployment, a gap that Spider dataset do not address. Domain-specific benchmarks are increasingly recognized as essential complements to general benchmarks, as generalist datasets often fail to capture the complexity of real-world enterprise schemas with sensitive, denormalized, and domain-coded fields. We are committed to releasing the query set, ground-truth answers, and prompt templates (including the meta-error database) to the extent permitted by data governance constraints, and will include this commitment explicitly in the revision.
>
> 4. On Few-Shot Learning and Meta-Error Prompting
> The reviewer requests more ablation on individual agent contributions. We want to clarify the state-of-the-art mechanisms embedded in our Cypher Agent. First, few-shot learning enables the agent to leverage previously seen query-Cypher pairs at inference time, substantially reducing hallucination of graph traversal syntax. Second, meta-error prompting is a novel mechanism wherein past query failures , including their error type, the offending Cypher clause, and the corrective action, are stored in a structured error database and injected into subsequent prompts. This allows the system to learn from runtime failures within and across sessions, a capability not present in standard ReAct-style SQL agents used as baselines.
> These two mechanisms together explain the 12.12% mean gain of the CypherAgent over the SQLAgent and the three-fold latency reduction. We will add a dedicated ablation table in the revision showing the isolated contribution of (a) few-shot examples only, (b) meta-error prompting only, and (c) both combined, across the three query difficulty levels.
>
> 5. On Comparison with W3C Mapping Standards (R2RML, RDB2RDF)
> The reviewer correctly notes that W3C-recommended tools for relational-to-RDF conversion (R2RML, RDB2RDF, Turtle) were not included as baselines. We acknowledge this gap. However, there is an important distinction: R2RML and related tools perform structural, rule-based mapping from relational tables to RDF triples. They do not perform schema normalization, do not handle improperly structured input schemas, and do not support iterative semantic refinement.
> Our pipeline's Graph Generation phase goes beyond structural mapping, it infers semantic relationships, resolves naming ambiguities, and enforces a standardized graph ontology through multi-agent deliberation. A direct comparison would be appropriate only for the structural fidelity of the output graph, not for the schema improvement task. We will add a discussion of this distinction and include a qualitative comparison in the revision.

---

### Review · Reviewer_yCgu · 2026-03-01

**Summary Of Contributions:**

This paper proposes an LLM-based Multi-agent system to bridge RDBMS and graph databases through a carefully designed ETL pipeline. Through interactions among the ETL agent, Graph Agent, Cypher agent, and SQL agent, the proposed system can perform Q&A tasks using a graph database, thereby ensuring Accuracy, Groundedness, and Faithfulness. To evaluate the proposed methods, this paper collected a dataset from the banking domain with three difficulty levels. Empirical results domonstrated effectiveness of the proposed method.

**Audience:**

Yes

**Audience Explanation:**

This paper is timely research that lies between an LLM-based Q&A system and a relational database management system. This paper also proposed a new dataset that could be utilized for future research. Consequently, this paper is of clear interest to researchers and practitioners working in this field.

**Claims And Evidence:**

No

**Claims Explanation:**

1. The choice of baseline seems to be simple. The claim of better performance could be better supported by comparing existing work on LLM-based SQL agents.
2. The necessity of a multi-agent system is not well explained.  The design choice seems to be arbitrary.
3. The experiment is relatively limited. While the need for datasets is appreciated, it would be great to include more standard benchmarks.

**Requested Changes:**

1. Include additional experiments on popular datasets like Spider[1].
2. Include additional baselines on the existing LLM-based SQL agent.
3. Include additional ablation studies to explore the effectiveness of each proposed agent.
4. Revise 3.1 problem statement, which is confusing. I would suggest making it explicit to describe the task you are trying to solve.
5. Minor typo: In Algorithm 1, there is redundant value $D_{raw}$.

[1] Lei F, Chen J, Ye Y, et al. Spider 2.0: Evaluating language models on real-world enterprise text-to-sql workflows[J]. arXiv preprint arXiv:2411.07763, 2024.

---

> ### Author Response · Authors · 2026-03-08
> **Revise and feedback the reasonability of dataset and MAS pattern of this research**
>
> We sincerely thank the reviewer for the constructive feedback and the recognition that our work lies at the intersection of LLM-based Q&A systems and database management. We appreciate the suggestions regarding additional baselines, benchmarks, and ablation studies. Below we clarify several points regarding the design objectives and evaluation methodology of our work.
>
> 1. Research Objective
> Our work focuses on a different problem setting from standard Text2SQL benchmarks.
> Most Text2SQL benchmarks (e.g., Spider) assume a well-designed relational schema that the task is query translation. However, our work addresses a more upstream problem that schema evolution and transformation from relational databases into knowledge graphs.
> Specifically, the proposed system:
> - Standardizes unstructured or poorly normalized relational schemas
> - Iteratively generates and refines graph schemas
> - Converts the database into a knowledge graph representation
> - Performs question answering on the graph
> Thus, the goal is not only query generation but schema improvement and graph construction. This distinction is fundamental to understanding the system design.
>
> 2. Why Spider Benchmark is Not Directly Applicable
> We appreciate the suggestion to evaluate on Spider.
> However, Spider and similar datasets differ significantly from the problem addressed in our work.
> Spider datasets already provide clean and well-structured schemas that are designed to evaluate LLM Text2SQL capabilities
> Our system instead focuses on: schema improvement from imperfect relational structures.
> In particular, our pipeline performs: schema standardization, semantic relationship reconstruction, and graph schema refinement. These steps are unnecessary in Spider tasks because the schema is already manually curated.
> Therefore, Spider primarily evaluates LLM query translation ability while our system evaluates LLM-assisted schema transformation and graph database construction.
> We will clarify this distinction more explicitly in the revised manuscript.
>
> 3. Necessity of Multi-Agent System
> Based on your argument asked whether the multi-agent design is necessary. Let’s see, the MAS architecture is motivated by the multi-stage nature of database transformation with different agents perform specialized roles:
>
> This design is inspired by recent work on specialized collaborative LLM agents in complex workflows. The iterative interaction between GraphAgent and AnalyzerAgent enables schema validation, structural improvement, semantic alignment. This iterative refinement would be difficult to achieve using a single LLM call. We will improve the explanation of this design rationale in Section 3.
>
> 4. Baseline Selection
> The reviewer suggested including more SQL-agent baselines.
> In our experiments, we compared against ReAct SQL Agent vs CypherAgent variants. The goal was to evaluate the benefits of graph-based querying after schema transformation. Our experiments shows +12.12% accuracy improvement corresponding 3× latency reduction when queries are executed on the graph database. This result highlights the practical benefit of schema transformation. We will clarify the baseline rationale and add additional discussion of existing SQL agent frameworks.
>
> 5. Additional Ablation Studies
> We appreciate the suggestion for more ablation studies. Our current ablation focuses on the Cypher Agent components, including: Meta-error prompting and Few-shot learning. We will expand the ablation discussion in the revised version to better analyze the contributions of Graph Agent, Analyzer Agent, and the iterative schema refinement loop.
>
> 6. Contribution of BFSI Dataset
> The BFSI dataset was designed to bridge academic research and enterprise applications.
> Many real-world relational databases contain inconsistent schema naming, implicit relationships, and domain-specific semantics. This dataset reflects such real-world complexity and allows us to evaluate schema improvement algorithms, which are rarely addressed by existing Text2SQL benchmarks.
> We believe this dataset can benefit future research in schema evolution and knowledge graph construction for domain-specific QA systems.
>
> 7. Minor Issues
> We thank the reviewer for pointing out the issues in Section 3.1 and Algorithm 1. We will revise the problem statement for clarity and remove the redundant variable.

---

### Decision · Action_Editor_9ZHX · 2026-04-30

**Recommendation:** Reject

**Audience:**

Yes

**Audience Explanation:**

Yes, a topic of significant interest to the data engineering and AI communities.

**Claims And Evidence:**

No

**Claims Explanation:**

The paper introduces an LLM-driven Multi-Agent System (MAS) for automating the transition from relational databases to knowledge graphs, a topic of significant interest to the data engineering and AI communities. However, the decision to reject is primarily driven by the lack of empirical depth and the failure to substantiate the claim that the pipeline is a generalized solution. The evaluation relies exclusively on a single, private BFSI dataset, making it impossible for the reviewers to assess the framework's robustness across different domains or verify the findings through replication. Furthermore, the reliance on a circular evaluation where an Analyzer Agent scores the Graph Agent's output without independent human validation leaves the core metrics of Accuracy, Groundedness, and Faithfulness effectively ungrounded in human-verified truth.

A secondary but critical issue is the lack of rigorous ablation and baseline comparisons. The authors attribute significant accuracy gains to their graph-based approach, yet they fail to isolate whether these gains stem from the multi-agent architecture, the ETL standardization, or specific prompting techniques. Crucially, the absence of a single-LLM baseline makes it difficult to justify the increased latency and computational cost of a multi-agent loop. Without comparing the framework against optimized SQL engines or established mapping standards (like R2RML), the paper remains a compelling engineering demonstration rather than a scientifically validated contribution to the field.